# Evaluation of nanopore sequencing for increasing accessibility of eDNA studies in biodiverse countries

Daniel Gygax[1,2,3]*, Sabina Ramirez[1], Moses Chibesa[4], Twakundine Simpamba[5], Michael Riffel[1], Tom Riffel[6], Amrita Srivathsan[7], Reindert Nijland[8], Lara Urban[2,3,9]*

1 Faunomics, Rifcon GmbH, Hirschberg, Germany, 2 Technical University of Munich, School of Life Sciences, Freising, Germany, 3 Helmholtz Munich, Neuherberg, Germany, 4 Department of Zoology and Aquatic Sciences, The Copperbelt University, Kitwe, Zambia, 5 Department of National Parks and Wildlife, Chilanga, Zambia, 6 Nsanga Conservation, Mfuwe, Zambia, 7 Center for Integrative Biodiversity Discovery, Museum für Naturkunde, Berlin, Germany, 8 Marine Animal Ecology Group, Wageningen University, Wageningen, The Netherlands, 9 Institute for Food Safety and Hygiene, University of Zürich, Zürich, Switzerland

* dgygax90@gmail.com (DG); lara.h.urban@gmail.com (LU)

## Abstract

Biodiversity loss is a global challenge of the 21st century. Environmental DNA (eDNA)-based metabarcoding offers a cost- and time-efficient alternative to conventional biodiversity surveys, enabling detection of rare, cryptic, and elusive species from environmental samples. However, limited access to genomic technologies restricts the application of eDNA metabarcoding in highly biodiverse remote regions and low- and middle-income countries (LMICs). Here, we directly compared the latest portable nanopore sequencing methods with established Illumina sequencing for vertebrate eDNA metabarcoding of Zambian water samples. Our results show that due to recent improvements in sequencing chemistry and optimized basecalling, nanopore sequencing data can recapitulate or even surpass established protocols, demonstrating the feasibility of *in situ* biodiversity assessments. eDNA- and camera trap-based species detections had minimal overlap in species detections, suggesting a complementary rather than substituting application of these biodiversity monitoring technologies. We finally demonstrate that our entire eDNA workflow can be successfully implemented in a mobile laboratory under remote field conditions by completing all steps—from sample collection to data analysis—within the Luambe National Park in Zambia. This approach has important implications for capacity building in LMICs and for overcoming limitations associated with sample export.

## Introduction

Biodiversity loss is one of the major challenges of the 21st century [1], but efforts to prevent biodiversity loss are often hindered by a lack of knowledge of species

**Data availability statement:** All sequencing data is deposited in the NCBI Sequence Read Archive (SRA; accession number: PRJNA1195924). The code used for this study is available 457 at: https://github.com/dmgr90/eDNA-Zambia-1.

**Funding:** The author(s) received no specific funding for this work.

**Competing interests:** Enter: The authors have declared that no competing interests exist.

distributions and dynamics [2,3]. Environmental DNA (eDNA) approaches can provide a cost- and time-efficient alternative to conventional biodiversity monitoring methods by detecting taxa, including rare, cryptic, and elusive species, from metabarcoding their genetic material in environmental samples such as water, soil, or air, and subsequent amplicon sequencing [4–6]. Such eDNA approaches have recently become cheaper and more reliable in monitoring the temporal and spatial distribution and dynamics of species, populations, and communities [7–9]. The lack of access to genomic technology, however, hinders the application of metabarcoding approaches in remote areas and low- and middle-income countries (LMICs), which often contain the globally largest amount of biodiversity [10–12]. Simultaneously, international treaties such as the Convention on International Trade in Endangered Species of Wild Fauna and Flora [13] and data sovereignty principles as specified by the Nagoya Protocol on Access and Benefit Sharing [14] and the Global Indigenous Data Alliance [15] often require the generation and usage of such genomic data close to the species' origin.

Long-read nanopore sequencing technology as developed by Oxford Nanopore Technologies (ONT) is the first genomic technology that enables rapid *in situ* sequencing at low upfront investment costs through its portable devices. This technology therefore holds the promise of putting eDNA research in line with international treaties and data sovereignty principles, while saving time and resources otherwise spent on sample transport and exportation, shortening the gap between research and decision-making and potentially empowering local researchers, conservationists, and decision-makers [11,16]. First applications of nanopore sequencing to metabarcoding in general [17–19] and eDNA specifically [20,21] have shown promising results in species but in comparison to the established sequencing platform for metabarcoding, short-read Illumina sequencing, remained affected by the relatively high sequencing error rates. Bludau *et al.* have, however, also shown that this lack in sequencing accuracy can be compensated by nanopore sequencing's long-read capability, which enabled species-level resolution in their protistan studies by substantially increasing the targeted amplicon size. While the error rate of this previously applied nanopore sequencing chemistry ("R9" chemistry) was estimated at anywhere between 5% and 22% [17,19], recent substantial improvements in the nanopore design ("R10" chemistry) and basecalling algorithms now allow for highly accurate nanopore read- and consensus-level accuracy. Doorenspleet *et al.* were the first to use these accuracy improvements to apply nanopore-based eDNA to successfully monitor marine vertebrates from aquarium and sea water. Similarly, Bludau *et al.* showed that an increase of the targeted amplicon size also improves species detection when using the latest nanopore chemistry; however, fragmentation of the marine vertebrate's DNA in real-world environments such as the sea inhibits the amplification of long stretches of DNA [22].

Here, we conducted an eDNA study to directly compare the latest improvements in portable nanopore sequencing with established Illumina sequencing as well as camera trapping for monitoring terrestrial vertebrate diversity from freshwater samples in Zambia. We additionally tested the *in situ* implementation of eDNA methods

in a mobile laboratory setting by applying our complete eDNA workflow from water collection to data analysis on site in Luambe National Park, Zambia. Zambia serves as a good example of a biodiversity-rich LMIC that shares borders and ecosystems with eight other African countries and is estimated to be home to 242 mammals, 757 bird, 74 amphibian and 156 reptiles [23]. Zambia has further been described as "a large patchwork of important ecosystems" [25]., with its sub-tropical savanna ecosystems being characterized by extreme oscillations of precipitation and water availability during the wet and dry seasons [24,25]. This seasonality results in an increased reliance on mobility for both humans and animals to access water [25] and therefore imposes challenges for monitoring highly mobile vertebrates across large areas. In arid and semi-arid systems where surface water is often scarce, natural and artificial water holes further act as gathering sites for mammals and birds [26], making them potentially optimal sources of eDNA for monitoring vertebrate fauna [27–29].

## Materials and methods

### Environmental DNA collection

For sequencing comparisons, we took water samples at five locations in the Luambe and Lukusuzi National Parks within the Luangwa Valley in Eastern Zambia between July 13th and 16th, 2023 (Fig A–F). In Luambe National Park, three water bodies were sampled: an artificial pond near an established tourism facility (CAM: 12.45880° S, 32.14662° E) (Fig 1B) and two permanent natural ponds (P2: 12.48503° S, 32.17606° E; P3: 12.48246° S, 32.19048° E) (Fig 1C, D). In Luku-suzi National Park, samples were collected from two sites along the Lukusuzi River (L1: 12.71192° S, 32.50254° E; L2: 12.71247° S, 32.50226° E) (Fig 1E, F). These locations were chosen because previous camera trap-based monitoring had

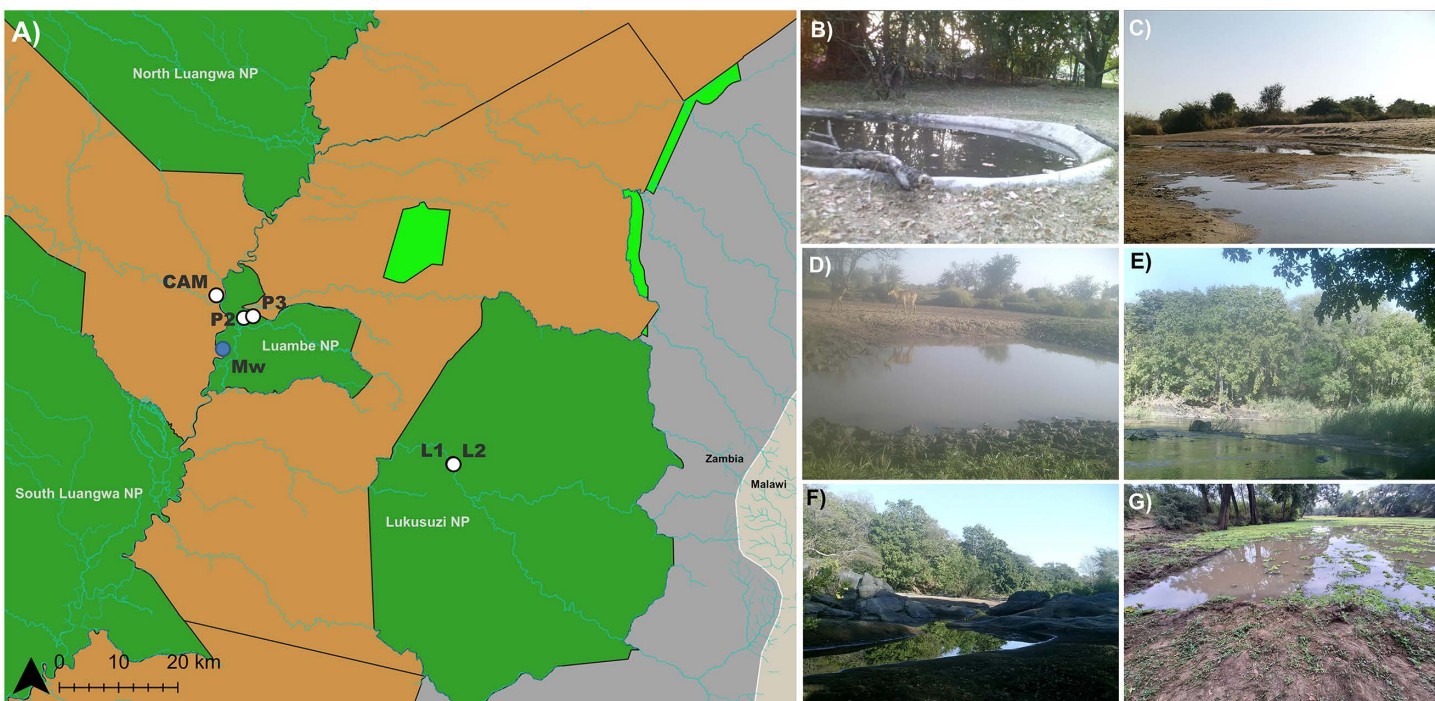

**Fig 1. Map and pictures of study sites.** A) Map showing the sampling locations within Luambe and Lukusuzi National Parks (Mw: blue dot); B) CAM: Artificial pond in Luambe Camp; C) P2: Natural waterhole 1 in Luambe NP; D) P3: Natural waterhole 2 in Luambe NP; E) L1: Lukuzusi backwater site 1; F) L2: Lukuzusi backwater site 2; G) Mw: Mwendi Lagoon (sampling site for the *in situ* application National Parks (green), Game Management Reserves (light brown), Forest Reserves (light green), and unprotected areas (grey) are shown. Map created by the authors using public source data © Open-StreetMap contributors (ODbL).

identified them as hotspots of vertebrate activity (Fig 1A–F). At each water body, three biological replicates (A, B, and C) were collected. For the *in situ* application (see "*In situ* eDNA application"), freshwater samples were collected at three sites of the Mwendi Lagoon (Mw: 12.53318° S, 32.1435° E) in Luambe National Park between June 11th and 16th, 2025 (Fig 1G).

We collected freshwater samples by using sterile plastic scoops and storing them in sterile Whirl-Pak® bags (Thermo Scientific, Germany) for transport to the camp. At the camp, one liter of water per sample was filtered using 250 ml cups with 0.45 µm nitrocellulose filters (Thermo Scientific Nalgene, Germany) and a Vampire Vacuum Pump (Bürkle, Germany) to speed up the process. For samples with high sediment content, multiple nitrocellulose filters were required to filter the entire liter due to clogging; for these samples, final DNA extracts were subsequently pooled across filters per replicate. After filtration, the nitrocellulose filters were first stored in sterile kraft bags and then placed in sterile ziplock bags with silica gel beads. The samples were frozen at -20ºC before and after transportation from Zambia to Germany. For the sequencing comparisons, all molecular work was subsequently conducted at the Helmholtz Center Munich laboratories in Germany under the corresponding research permits from the Zambian Department of National Parks and Wildlife (Reference number: NPW/8/27/1).

## Environmental DNA extraction

The nitrocellulose filters were cut in half using a sterile scalpel, sterile gloves, and a FFP2 mask, then stored separately in 1.5 ml Eppendorf tubes at −20°C until further processing; half of a filter was then used for DNA extraction while the other half was kept as a backup. DNA was extracted from all samples using the DNeasy PowerSoil Kit (Qiagen) following the manufacturer's instructions, with the exception that DNA was eluted in only 50 µl of elution buffer in the final step.

For the filter negative control, one liter of sterile Milli-Q® water was filtered. Together with all biological replicates across all sampling sites, this resulted in 16 DNA extracts, whose DNA concentrations were measured with an Invitrogen™ Qubit™ 3.0 Fluorometer using the High Sensitivity (HS) Assay.

## PCR amplification and metabarcoding

Metabarcoding was performed using two mitochondrial primer sets. The vertebrate-specific 12S rRNA primer pair 12SV05 (forward 5′-TTAGATACCCC ACTATGC-3′; reverse 5′-TAGAACAGGCT CCTCTAG-3′) amplifies a∼97 bp fragment of the 12S rRNA gene [30]. Mammals were further targeted with the 16S rRNA primer pair 16Smam1 (forward 5′-CGGTTGG GGTGACCTCGGA-3′) and 16Smam2 (reverse 5′-GCTGTTATCCCTAGGGTAACT-3′), which amplifies a∼95 bp fragment of the 16S rRNA gene [31].

Each sample extract was amplified in PCR triplicates, and a filter negative control in two PCR replicates, plus one PCR control, resulting in overall 48 PCR products. The PCR reactions were set up in 25 µL volumes consisting of 2 µL DNA template, 12.5 µL Phusion® High-Fidelity PCR Master Mix, 1.25 µL of each primer, and 8 µL of DNase-free water. For the 12S rRNA primer pair the cycling parameters were as follows: 95°C for 10 minutes, followed by 40 cycles of 94°C for 30 s, 60°C for 30 s, and 70°C for 60 s, with a final extension at 72°C for 7 min. For the 16S rRNA primer pair, the cycling parameters were as follows: 95°C for 10 min, followed by 40 cycles of 95°C for 12 s, 59°C for 30 s and 70°C for 25 s, with a final extension at 72 °C for 7 min.

A symmetrical tagged primer approach was used to multiplex the samples during amplification. Unique 9 bp tags, generated using Barcode Generator (https://github.com/lcomai/barcode_generator), were added to the 5′ ends of both the forward and reverse primers for each primer set. This tag length has been shown to be suitable for multiplexing amplicons for nanopore sequencing [32]. All tags were designed to differ by at least 3 bp from each other, and the same tag was used on both the forward and reverse primers of each PCR product.

Following amplification, one replicate PCR product was respectively visualized on 2% agarose gels with GelRed and Invitrogen™ 100 bp DNA. Amplification was confirmed by the presence of a band of the expected size, which was the case for all visualized samples (including the negative controls).

## Sequencing library preparation and sequencing

The 48 PCR products were pooled by taking 10 µl of each product and then cleaned using a 2:1 bead-to-DNA ratio with AMPure XP Beads (Beckman Coulter, Germany), followed by resuspension in 100 µL of DNase-free water. This final pool was separated into two 50 µL volumes as input for nanopore and Illumina sequencing, respectively.

Nanopore sequencing was performed using the portable MinION Mk1B device. Library preparation was done using the Ligation Sequencing Kit (SQK-LSK114), following the manufacturer's Ligation Sequencing Kit Amplicons protocol with the following modification: At the end of the protocol, 30 µL of elution buffer was used instead of the specified 15 µL. This was done to simultaneously prepare two libraries, which was possible due to high DNA input into the library preparation. Both libraries were consecutively sequenced for 16 hours on one flow cell, separated by a flow cell wash step.

Illumina sequencing was performed by Novogene, GmbH, Germany, using their 2x150 bp protocol on a NovaSeq 6000.

## Data processing and OTU clustering

Nanopore raw data was basecalled using Dorado's (v1.0.2) super high accuracy model (SUP v5.0.0; https://github.com/nanoporetech/dorado). The resulting FASTQ files were demultiplexed using OBITools4's obimultiplex command, allowing for a maximum of two errors in the tags (https://git.metabarcoding.org/obitools/obitools4). Primers were trimmed using Cutadapt (v4.9) [33]. Subsequent Operational Taxonomic Unit (OTU) processing steps were performed using VSEARCH (v2.28.1) [33]: We first filtered the reads by setting the VSEARCH fastq_maxee parameter to thresholds of 1.0: For a 97 bp sequence, a fastq_maxee value of 1.0 corresponds to average per-base quality scores of approximately 20 using the formula: $Q = -10 \times \log_{10}(\text{fastq\_maxee} \div L)$, where Q is the average per-base Phred quality score and L is the read length. Therefore, lower fastq_maxee thresholds require higher average per-base quality scores to keep the expected error per read below the specified limit. The next steps included dereplication (with singleton removal), chimera removal, and clustering at a 99% sequence identity threshold; the final OTUs were pooled across respective PCR replicates.

Illumina paired-end data were first merged and then demultiplexed with Obitools 4 allowing for a maximum of two errors in the tags. Primers were trimmed using Cutadapt (v4.9) [33]. Subsequent OTU processing steps were identical to the OTU processing steps of the nanopore sequencing data.

## Taxonomic assignments

Amphibian, bird, mammalian, and bird species lists reported in Zambia were obtained from the SASCAL portal (http://data.sasscal.org/metadata/view.php?view=doc_documents&id=3154, 2017). This full species list (S1 Table) was used as a list of regional species, and to assess our study's database incompleteness database (S2 Table).

Taxonomic assignment of the OTUs was done using the global alignment function usearch_global VSEARCH (v2.28.1) [34] against the Midori2 databases Unique Nucleotide short ribosomal RNA version 266 (MIDORI2_UNIQ_NUC_GB266_srRNA_BLAST) and the Unique Nucleotide long ribosomal RNA version 266 (MIDORI2_UNIQ_NUC_GB266_lrRNA_BLAST), which contain curated non-redundant 12S and 16S rRNA sequences, respectively [35]. Only the best taxonomic hits with at least 90% sequence similarity score and 80% query coverage score were taken and then filtered for local taxa detections based on the list of regional species: Assignments with ≥98% identity to a reference sequence from Zambia were considered reliable species-level identifications: matches between 95–98% identity were assigned at the genus level, while those between 90–95% identity were conservatively classified at the family level. We only focused our subsequent analyses on primary non-terrestrial vertebrate classes. Taxa detected in the negative controls were removed from the data.

## Camera trapping

One week prior to eDNA sampling, camera traps (SECACAM Wild-Vision Full HD 5.0) were put up at sampling locations to validate species detections. Cameras, equipped with passive infrared sensors, were set to picture-mode and recorded for 24

hours. The number of deployed camera traps varied from 2 to 5 cameras across sites, depending on the size and accessibility of the water bodies. Cameras were positioned to cover the entire waterbodies with priority on established drinking places by large mammals. Drinking places were determined by visual assessments on mammal tracks, scat, or further signs of presence.

### *In situ* eDNA application

We finally conducted our entire protocol including sample collection, molecular work, and bioinformatic analyses on site in the Nsanga Research Camp within Luambe National Park. Freshwater samples were collected daily from three sites (A–C) spaced ~30 m apart at the Mwendi Lagoon (Mw: 12.53318° S, 32.1435° E) in Luambe National Park from June 11th to 16th. For the molecular work, we used a mobile laboratory setting assembled within a camping tent.For the bioinformatic analyses, we used a high-performing laptop (Razer Blade 15.6″ with 16 GB RAM, an NVIDIA RTX 4070 GPU, and an Intel Core i7 processor). We again used camera trapping in parallel (see "Camera Trapping"), with one camera per sampling site and daily assessments from sampling day 2.

We implemented the following modifications of our nanopore sequencing-based eDNA protocol for the *in situ* approach:

1) To reduce contamination risk, the following protocol steps were modified: Instead of using 0.45 μm nitrocellulose filters and centralized processing by a vacuum pump in the camp, Smith and Root™ eDNA Filter Packs were used to filter three liters of wateron site while minimizing handling. We used PES (Polyethersulfone) filters with a larger pore size (1.2 μm) to reduce clogging when filtering larger volumes of water. With this approach it took between 15 min and 25 minutes to filter 3L of water. The filters were stored in sterile bags and stored at -20°C until DNA extraction.

2) We used the DNeasyPowerWater Kit (Qiagen) for DNA extraction following the manufacturer's instructions, with the exception that DNA was eluted in 50 μl of elution buffer in the final step. We used this kit instead of the DNeasy PowerSoil Kit (Qiagen) since it allows to process the entire filter in a faster manner. For the filter negative controls, we used blank filters instead of filtering sterile Milli-Q® water due to lack of access to such volumes of sterile water. We used one blank filter for each of the three DNA extractions of six samples. This resulted in 21 DNA extracts, whose DNA concentrations were measured with an onsite Invitrogen™ Qubit™ 3.0 Fluorometer using the High Sensitivity (HS) Assay.

3) We incorporated human blockers in the PCR reaction to reduce human contamination. The PCR reactions changed but the same symmetrical tagging approach as described previously was used for sample multiplexing. Specifically, the PCR reactions were set up in 20 μL volumes consisting of 3 μL DNA template, 0.75 U AmpliTaq Gold, 1 × Gold PCR Buffer and 2.5 mM $MgCl_2$ (all reagents from Applied Biosystems); 0.6 μM each of 5′ nucleotide tagged forward and reverse primer; 0.2 mM dNTP mix (Invitrogen); 0.5 mg/mL bovine serum albumin (BSA, Bio Labs), and 3 μM human blocker (5′–3′ TACCCCACTATGCTTAGCCCTAAACCTCAACAGTTAAATC–spacerC3 for the 12S rRNA primers [36] and 5′–3′ GCGACCTCGGAGCAGAACCC–spacerC3 for the 16S rRNA primerss [37]). For the 12S rRNA PCR protocol, the number of cycles was increased from 40 to 45 cycles, and instead of the 60°C we used 59°C. For the 16S rRNA PCR protocol, instead of the 59 °C we used 51 °C. We confirmed amplification using 2% EX pre-cast agarose gels run with the E-Gel™ Power Snap Electrophoresis (Thermo Fischer Scientific). For the PCR pool clean-up, we further changed the bead:sample ratio from 2:1 to 1.5:1 to reduce primer-dimer carryover.

4) Nanopore sequencing was performed using the portable MinION Mk1D device with better temperature control than the Mk1B device.

## Results

Read counts varied widely across samples for both sequencing technologies and primer set. For Illumina 12S samples ranged from 3 to 161,352 reads per sample (mean = 48,459; SD = 41,615). For Illumina 16S yields ranged from 448 to

470,489 (mean = 121,653; SD = 89,537). For Nanopore 12S yields spanned 18–2,424,297 (mean = 273,710; SD = 543,585). For Nanopore 16S yields ranged from 2,029–1,433,579 (mean = 227,434; SD = 309,771).

Our database revision revealed that the availability of genetic references for Zambian biodiversity in the NCBI repository was limited across all primary terrestrial vertebrate classes 58.14% of occurring amphibian species were covered by 12S rRNA entries, 68.60% by 16S rRNA, but only 6.98% by full mitochondrial references. 42.78% of reptile species were covered by 12S rRNA entries, 64.43% by 16S rRNA, but only 3.09% by full mitochondrial references. 28.38% of avian species were covered by 12S rRNA entries, 16.43% by 16S rRNA, and 15.64% by full mitochondrial references. Finally, mammals were relatively well represented with 58.80% (12S rRNA), 53.60% (16S rRNA) and 57.60% (full mitochondria), respectively.

All four primary terrestrial vertebrate classes were detected by nanopore and Illumina sequencing; as *Xenopus* sp. was detected by negative controls, it was excluded from downstream analyses. For both sequencing technologies the highest number of local species detections occurred for mammals (17 and 11 species, respectively; Fig 2). Altogether, 18 taxa were detected by both sequencing technologies while seven taxa were only detected by nanopore, and one taxon only by Illumina sequencing (Fig 2). The two primer sets showed little overlap in amplified taxa, with the 12s rRNA primer set yielding 21 taxa and the 16s rRNA primers 9 taxa, with only four of them in common. Most taxa were detected at the CAM site (n = 15) followed by P3 and L1 (n = 7, respectively), and L2 (n = 6) and P2 (n = 5). While most sites showed little overlap in detected taxa, a few common species like *Hippopotamus amphibius (*Hippopotamus*)*, *Chlorocebus pygerythrus* (Vervet monkey) and *Papio cynocephalus* (Yellow baboon) were detected at nearly all sites (Fig 2).

We next compared all our taxa detections between the different sequencing technologies and camera trapping (Fig 3). Briefly, camera trapping-based detection was biased towards birds and mammals with 10 and 9 species detected, respectively, and no detections for amphibians or reptiles (S3 Table) while eDNA-based methods yielded 25 detections across all four primary terrestrial vertebrates, out of which 14 were at the species level and 5 at the genus and 7 at the family level, respectively. We observed little overlap between the taxa detected by camera traps and eDNA; this overlap increased slightly with taxonomic ranks (Fig 3).

We finally validated a full mobile laboratory workflow based on nanopore sequencing under field conditions (Methods). Over six consecutive sampling days at three sites of Mwendi Lagoon, we detected ten primary terrestrial vertebrate taxa using eDNA, and seven with camera traps, with four of the taxa detected by both methods (Fig 4A). Within the eDNA detections, nine taxa were recovered with the 12S rRNA primer, and four with the 16S rRNA primer, including three taxa amplified by both primers (Fig 4B). No local species was detected in the negative controls. Taxonomic richness varied across the three sites: Site A yielded the highest richness by eDNA approaches (n = 9), followed by sites B and C (n = 4 and n = 3, respectively); more taxa could be detected by eDNA than by camera trapping at A and C, and *vice versa* at B (Fig 4C).

We found that frequent visitors (e.g*., Loxodonta africana*, *Papio cynocephalus*) and semi-aquatic taxa (e.g., *H. amphibius*, *Xenopus sp*.) were detected on multiple days and by more than one approach, whereas birds, bats, and rodents were typically recorded on a single day and by only one method. All antelopes—*Aepyceros melampus, Tragelaphus scriptus, Kobus vardonii*, and *Kobus ellipsiprymnus*—were detected exclusively by camera trapping and were not recovered from eDNA samples (Fig 5).

## Discussion

While eDNA approaches hold promise for providing a cost- and time-efficient alternative to conventional biodiversity monitoring approaches such as camera trapping, the *in situ* application of eDNA in biodiverse areas and countries has so far been hampered by the reliance of standard protocols on Illumina sequencing technology, which requires expensive machinery that is mostly only available at centralized sequencing facilities or research laboratories [4,5,21]. It is, however, important to conduct eDNA analyses on site—to obtain results without delay, empower local decision-makers, and to fulfill

| Taxonomic ranks | | | | Sampling sites | | | | |
|---|---|---|---|---|---|---|---|---|
| Class | Order | Family | Species | CAM | P2 | P3 | L1 | L2 |
| Amphibians | Anura | Ptychadenidae | *Ptychadena anchietae* | ▲ | | | | |
| Birds | Charadriiformes | Charadriidae | *Vanellus armatus* | | ● | | | |
| | Anseriformes | Anatidae | *Anas sp.* | | | | | ● |
| | Bucerotiformes | Bucorvidae | *Bucorvidae sp.* | ● | | | | |
| | Cuculiformes | Cuculidae | *Cuculidae sp.* | ● | | | | |
| | Accipitriformes | Accipitridae | *Accipitridae sp.* | ● | | | | |
| | Psittaciformes | Psittacidae | *Poicephalus sp.* | ● | | | | |
| | Anseriformes | Anatidae | *Sarkidiornis melanotos* | ▲ | | | | |
| Mammals | Artiodactyla | Hippopotamidae | *Hippopotamus amphibius* | ●▲ | ●▲ | ●▲ | ●▲ | ●▲ |
| | Artiodactyla | Bovidae | *Syncerus caffer* | | ▲ | | | |
| | Artiodactyla | Bovidae | *Tragelaphus sp.* | ● | | ● | | ● |
| | Artiodactyla | Bovidae | *Tragelaphus scriptus* | | | ▲ | | |
| | Carnivora | Herpestidae | *Helogale parvula* | | | ●▲ | ●▲ | |
| | Carnivora | Felidae | *Felidae sp.* | ● | | ● | ● | |
| | Carnivora | Herpestidae | *Herpestes sp.* | | | | ● | |
| | Carnivora | Viverridae | *Viverridae sp.* | | | | ● | |
| | Chiroptera | Vespertilionidae | *Afronycteris nanus* | ●▲ | | | | |
| | Chiroptera | Vespertilionidae | *Vespertilionidae sp.* | ▲ | | | | |
| | Chiroptera | Rhinolophidae | *Rhinolophus sp.* | ● | | ● | | |
| | Primates | Cercopithecidae | *Chlorocebus pygerythrus* | ●▲ | ●▲ | ●▲ | ●▲ | |
| | Primates | Cercopithecidae | *Papio cynocephalus* | ▲ | ▲ | | ▲ | ▲ |
| | Rodentia | Muridae | *Muridae sp.* | | ● | | | |
| | Rodentia | Gliridae | *Graphiurus kelleni* | | | | | ● |
| | Rodentia | Gliridae | *Graphiurus microtis* | | | | | ● |
| | Rodentia | Muridae | *Pelomys fallax* | ▲ | | | | |
| Reptiles | Testudines | Pelomedusidae | *Pelusios sinuatus* | | | | | ● |

| Legend | | |
|---|---|---|
| Marker | 12S | ● |
| | 16S | ▲ |
| | Both | ●▲ |
| Seq. tech | Nanopore | (blue) |
| | Illumina | (red) |
| | Both | (pink) |

**Fig 2. Primary terrestrial vertebrate taxa detected.** Detections are depicted across the five Zambian sampling sites eDNA primers sets (12S or 16S rRNA, or both) and sequencing technology (nanopore and Illumina sequencing, or both).

international treaties on data sovereignty, material export, and benefit sharing when it comes to genetics-based biodiversity monitoring [14,15].

Here, we established portable nanopore sequencing for eDNA analysis of freshwater samples from Zambia for the detection of primary terrestrial vertebrate species and then compared established Illumina and nanopore sequencing-based eDNA results with each other based on the same DNA extracts and the same targeted marker gene regions. We show that the improvements in nanopore sequencing accuracy through sequencing chemistry updates (R10.4.1) and improved data basecalling algorithms (Dorado SUP) lead to mostly comparable species detections between the two eDNA approaches, with nanopore sequencing even detecting more local species than Illumina sequencing. Based on these results, we applied a fully mobile approach based on the portable nanopore genomic technology and cost-effective sample multiplexing for *in situ* biodiversity monitoring on site in the Luambe National Park in Zambia, confirming the *in situ* applicability of our eDNA approach.

Due to its long sequencing reads, nanopore sequencing further holds promise for extending future metabarcoding studies to longer marker gene regions, or entire mitochondrial genomes [18,21,22,38]. For example, Bludau *et al.* showed that nanopore-sequenced amplicons of the full-length 18S rRNA gene from soil protistan communities achieved greater taxonomic classification accuracy down to the species level than short-read Illumina-sequenced amplicons of the Variable

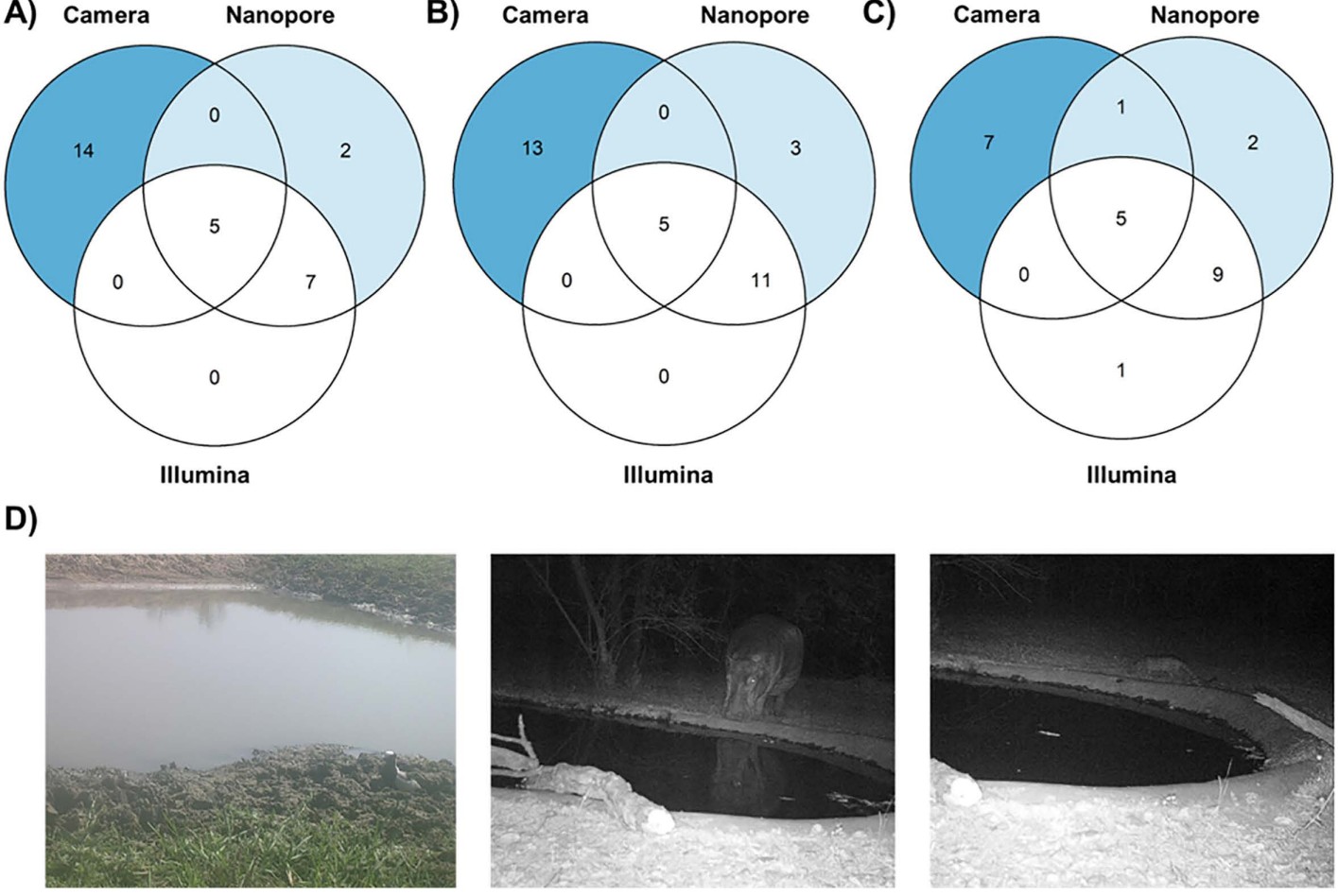

**Fig 3. Shared taxa detections between camera trap-based biodiversity monitoring and freshwater-based eDNA-based monitoring using nanopore and Illumina sequencing.** Taxonomic A) species, B) genus, and C) family detections. D) Examples of camera trap pictures: *left*: Blacksmith lapwing (*Vanellus armatus*) detected by all three approaches; *middle*: Hippopotamus (*Hippopotamus amphibius*) detected by all three approaches; and *right*: Small-spotted Genet (*Genetta genetta*) detected by camera trapping and by nanopore sequencing on the family level (*Viverridae sp.).*

Region 9 of the same gene, which only provided reliable classification on the phylum level [18]. Similarly, Doorenspleet, *et al.* compared nanopore-sequenced amplicons of a 2 kb region of fish mitochondrial DNA against the commonly used MiFish primer pair targeting a ~ 170 bp region. They showed that while the longer amplicons allowed for more species assignment in a controlled aquarium condition, the shorter amplicons performed better in natural environmental settings. This is probably because eDNA is often highly fragmented due to various environmental factors such as UV radiation, microbial activity, and enzymatic degradation, resulting in higher persistence time for shorter fragments [39–41]; such shorter DNA fragments are furthermore likely to amplify during PCR reactions [42]. When using eDNA to monitor vertebrates, especially primary terrestrial vertebrates from freshwater samples like presented in our study, the samples' DNA might therefore be too fragmented to allow for efficient long-read gene sequencing, which remains to be assessed. The suitability of amplicon length in metabarcoding studies might ultimately depend on different factors, such as the targeted taxa, the eDNA source, and the amount of genetic variability contained within the amplified marker. In highly diverse regions (e.g., the Luangwa Valley) that host several sympatric closely related species, for example among antelopes and felids, the use of longer amplicons could provide the taxonomic resolution needed to discriminate among species.

PLOS One | https://doi.org/10.1371/journal.pone.0333994   October 16, 2025                                          9 / 14

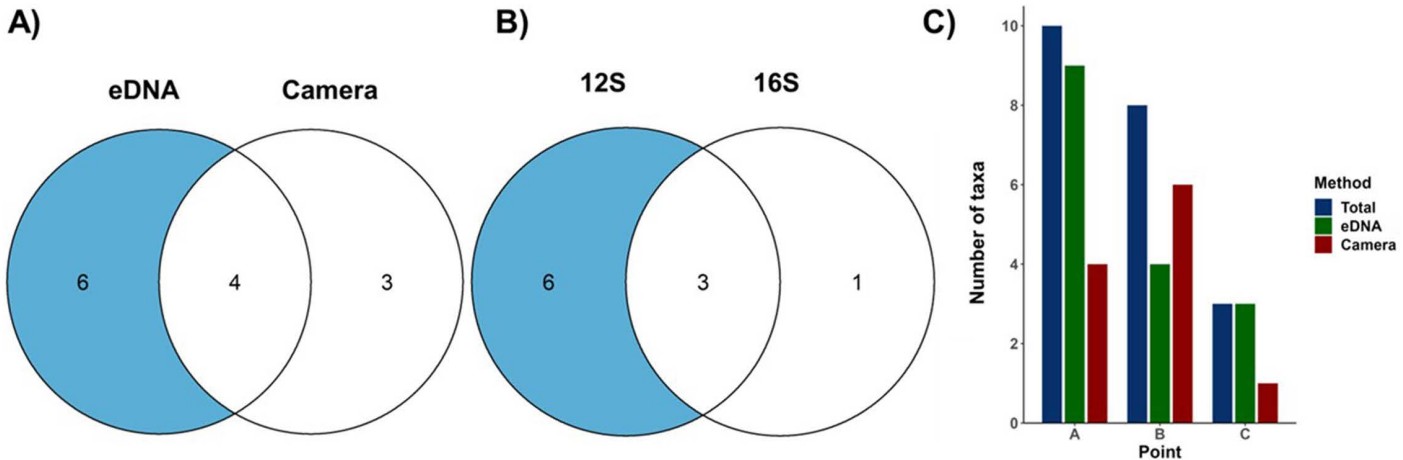

**Fig 4. Number of primary terrestrial vertebrate taxa accumulated over six days detected.** A) *in situ* eDNA and camera trapping; B) the two different primers used for *in situ* eDNA (12S and 16S rRNA); and C) by all approaches across the three sampling sites at the Mwendi Lagoon, Zambia.

Based on our database exploration, we found that, even for small marker gene regions (such as the 12S and 16S rRNA regions used in this study), genomic databases remain incomplete, particularly for megadiverse but understudied ecosystems [42–44]—such as Zambia's subtropical savannahs ecosystems as assessed in this study. The accurate mapping of marker gene OTUs to publicly available databases such as the entire NCBI nucleotide or the MIDORI2 databases strongly depends on the regional taxonomic coverage and can lead to unassigned sequences or false-positive assignments in the case of lacking references [45–47]. We therefore emphasize that building or populating genetic databases with regional genetic diversity is crucial for the successful application of eDNA based monitoring approaches; until databases are more complete, any eDNA application must be interpreted while taking the availability of genetic information into account.

We further compared our eDNA approaches with more conventional biodiversity monitoring achieved through simultaneously applied camera trapping Both our eDNA approaches showed little overlap with camera-based species detections. Several methodological and ecological factors could explain the discrepancies. As discussed previously, database completeness, primer choice, and amplicon resolution can influence which taxa are detected by eDNA. In addition, the interaction between species and the eDNA source can determine the likelihood of detection. For example, three recent studies have shown that waterborne eDNA from African waterholes can successfully reveal mammalian and other vertebrate communities [27–29]. However, we argue that these methods are likely to perform better for certain species than others, depending on their ecology. Water-dwelling or drought-intolerant species, such as many mammal or bird species, are more likely to leave detectable DNA traces in such waterholes. eDNA persistence in waterholes has further shown to be relatively short (Farell, 2022), which can contribute to variability in detection rates among species. Our findings illustrate this ecological bias, with eDNA preferentially detecting species with frequent or prolonged contact with water, such as hippos, elephants, and amphibians. Species that interact only briefly with water bodies, such as antelopes and felids, may not shed sufficient DNA to be consistently detected by eDNA, but could reliably be detected by camera trapping. Similarly, drought-resistant species or those able to obtain water from food resources may leave little or no trace in water samples. On the other hand, camera traps might miss species that cannot be easily detected visually, such as, for example, reptiles, amphibians, and small mammals such as rodents, bats, and pangolins. Overall, the limited overlap between eDNA and camera trapping suggests their complementary application and highlights the value of integrating additional eDNA sources such as air, soil, or blood-feeding insects [48–52].

| Class | Order | Species | Method | Day 1 | Day 2 | Day 3 | Day 4 | Day 5 | Day 6 |
|---|---|---|---|---|---|---|---|---|---|
| Amphibians | Anura | *Hyperolius sp.* | eDNA | | | | | ● | ● |
| | | | Camera traps | NA | | | | | |
| | Anura | *Xenopus sp.* | eDNA | | | | | ● | |
| | | | Camera traps | NA | | | | | |
| Birds | Galliformes | *Numida meleagris* | eDNA | | | ● | | | |
| | | | Camera traps | NA | | | | | |
| | Pelecaniformes | *Threskiornis aethiopicus* | eDNA | | | | | | ● |
| | | | Camera traps | NA | | | | | |
| Mammals | Artiodactyla | *Hippopotamus amphibius* | eDNA | ●▲ | ●▲ | ●▲ | ●▲ | ●▲ | ●▲ |
| | | | Camera traps | NA | | | | | |
| | Artiodactyla | *Aepyceros melampus* | eDNA | | | | | | |
| | | | Camera traps | NA | | | | | |
| | Primates | *Papio cynocephalus* | eDNA | | | ● | | ● | |
| | | | Camera traps | NA | | | | | |
| | Chiroptera | *Epomophorus pusillus* | eDNA | | | | ● | | |
| | | | Camera traps | NA | | | | | |
| | Rodentia | *Thallomys paedulcus* | eDNA | | | | | | ▲ |
| | | | Camera traps | NA | | | | | |
| | Proboscidea | *Loxodonta africana* | eDNA | ▲ | ▲ | ▲ | ▲ | | |
| | | | Camera traps | NA | | | | | |
| Amphibians | Anura | *Xenopus sp.* | eDNA | ● | | ● | ● | ● | ● |
| | | | Camera traps | NA | | | | | |
| Birds | Galliformes | *Numida meleagris* | eDNA | | | | | | |
| | | | Camera traps | NA | | | | | |
| | Passeriformes | *Vidua macroura* | eDNA | | | | ● | | |
| | | | Camera traps | NA | | | | | |
| Mammals | Artiodactyla | *Hippopotamus amphibius* | eDNA | ●▲ | ●▲ | ●▲ | ●▲ | ▲ | ●▲ |
| | | | Camera traps | NA | | | | | |
| | Artiodactyla | *Kobus vardonii* | eDNA | | | | | | |
| | | | Camera traps | NA | | | | | |
| | Artiodactyla | *Tragelaphus scriptus* | eDNA | | | | | | |
| | | | Camera traps | NA | | | | | |
| | Artiodactyla | *Kobus ellipsiprymnus* | eDNA | | | | | | |
| | | | Camera traps | NA | | | | | |
| | Primates | *Papio cynocephalus* | eDNA | ●▲ | ●▲ | | ●▲ | ●▲ | |
| | | | Camera traps | NA | | | | | |
| Amphibians | Anura | *Xenopus sp.* | eDNA | ● | ● | ● | ● | ● | ● |
| | | | Camera traps | NA | | | | | |
| Mammals | Artiodactyla | *Hippopotamus amphibius* | eDNA | ●▲ | ●▲ | ●▲ | ●▲ | ●▲ | ●▲ |
| | | | Camera traps | NA | | | | | |
| | Proboscidea | *Loxodonta africana* | eDNA | ●▲ | ▲ | ▲ | | | |
| | | | Camera traps | NA | | | | | |

**Legend**

| Point | A (blue) |
|---|---|
| | B (orange) |
| | C (green) |
| Marker | 12S ● |
| | 16S ▲ |
| | Both ●▲ |

**Fig 5. Primary terrestrial vertebrate taxa detected by *in situ* eDNA and camera trapping.** Detections are shown for three sampling point (A, B, C) at the Mwendi Lagoon, Zambia, across six consecutive sampling days (Methods). For eDNA detections, the primers 12S (●) and 16S (▲) rRNA are indicated.

In summary, we show that nanopore sequencing technology can be used to reliably carry out eDNA-based biodiversity monitoring in remote locations, including in highly diverse LMICs such as Zambia. The mobile laboratory and protocols we describe can be easily adapted for processing other eDNA sources, or for other conservation purposes such as pathogen

surveillance. These efforts can pave the way towards democratization of genomic technology in general, and eDNA applications specifically. However, further efforts are needed to (i) make eDNA-based approaches more cost-effective so that capacity can be built for routine applications in LMICs, particularly for conservation purposes where resources are often limited, and to (ii) contribute to regional reference database developments to better reflect global genetic biodiversity.

## Supporting information

**S1 Table. List of primary terrestrial vertebrates reported in Zambia.** List of mammal, bird, amphibian, and reptile species reported from Zambia, compiled from the sources cited in the Methods section.
(TXT)

**S2 Table. Zambian terrestrial vertebrates with or without 12S, 16S, or mitochondrial references.** Species from the Zambia terrestrial vertebrate list (S1 Table), indicating whether reference sequences for 12S, 16S, or complete mitochondrial genomes are available in NCBI.
(TXT)

**S3 Table. Species identified by camera traps across the five sample sites.** Species of primary terrestrial vertebrates recorded by camera traps at the five sampling sites (CAM, P2, P3, L1, and L2).
(TXT)

## Author contributions

**Conceptualization:** Daniel Gygax, Sabina Ramirez, Michael Riffel, Lara Urban.

**Data curation:** Daniel Gygax, Amrita Srivathsan, Reindert Nijland.

**Formal analysis:** Daniel Gygax.

**Methodology:** Daniel Gygax, Moses Chibesa, Twakundine Simpamba, Tom Riffel, Amrita Srivathsan, Reindert Nijland, Lara Urban.

**Project administration:** Sabina Ramirez, Twakundine Simpamba, Michael Riffel.

**Resources:** Moses Chibesa, Twakundine Simpamba, Michael Riffel, Tom Riffel.

**Software:** Daniel Gygax, Amrita Srivathsan, Reindert Nijland.

**Supervision:** Sabina Ramirez, Lara Urban.

**Visualization:** Daniel Gygax, Amrita Srivathsan, Lara Urban.

**Writing – original draft:** Daniel Gygax, Lara Urban.

**Writing – review & editing:** Tom Riffel, Amrita Srivathsan, Reindert Nijland, Lara Urban.

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
