## [Decision Letter · Decision Letter 0]

8 Jul 2025

Dear Dr. Gygax,

Thank you for submitting your manuscript to PLOS ONE. After careful consideration, we feel that it has merit but does not fully meet PLOS ONE’s publication criteria as it currently stands. Therefore, we invite you to submit a revised version of the manuscript that addresses the points raised during the review process.

**ACADEMIC EDITOR:** The manuscript provides useful findings. The authors are highly encourage to pay attention to meet reviewers comments successfully.

We look forward to receiving your revised manuscript.

Kind regards,

Mizanur Rahman, Ph.D.

Academic Editor

PLOS ONE

Journal Requirements:

1. Please ensure that your manuscript meets PLOS ONE's style requirements, including those for file naming. The PLOS ONE style templates can be found at https://journals.plos.org/plosone/s/file?id=wjVg/PLOSOne_formatting_sample_main_body.pdf and https://journals.plos.org/plosone/s/file?id=ba62/PLOSOne_formatting_sample_title_authors_affiliations.pdf.

2.  Please include your tables as part of your main manuscript and remove the individual files. Please note that supplementary tables (should remain/ be uploaded) as separate ""supporting information"" files".

Reviewers' comments:

Reviewer's Responses to Questions

**Comments to the Author**

1. Is the manuscript technically sound, and do the data support the conclusions?

Reviewer #1: Yes

Reviewer #2: Partly

Reviewer #3: Partly

Reviewer #4: Yes

2. Has the statistical analysis been performed appropriately and rigorously?

Reviewer #1: Yes

Reviewer #2: I Don't Know

Reviewer #3: Yes

Reviewer #4: Yes

3. Have the authors made all data underlying the findings in their manuscript fully available?

Reviewer #1: Yes

Reviewer #2: Yes

Reviewer #3: Yes

Reviewer #4: Yes

4. Is the manuscript presented in an intelligible fashion and written in standard English?

Reviewer #1: Yes

Reviewer #2: Yes

Reviewer #3: Yes

Reviewer #4: Yes

Reviewer #1: Current manuscript is well defined and satisfactory, however I would suggest some minor changes as required to be considered below,

Abstract satisfactory

Introduction satisfactory

Methodology satisfactory, but must add the citation for defining the methodology adopted by you for this article

Results and discussion well defined and satisfactory, but add the forward and reverse sequence of each species identified by you and compare them with 5 previously identified species from NCBI data base as you mentioned in your methodology for comparison between your eDNA technique with them. Define the percentage of matching with previous barcoding of each species with your data.

If possibility compare your major outcomes of usage of eDNA technology for vertebrates species with other dna barcoding technologies.

Also add the morphological identification characters of your each identify species and compare it with previous publication, so middle income countries researchers also used them for their regional identification of fauna found in their freshwater resources

Wish you best of luck

Regards

A reviewer

Reviewer #2: The study by Dr Gygax et al., is interesting in that it focuses on a modern theme whose application and accuracy still require years of work before being fully effective. Therefore, any exploratory contribution in this area is essential.

The study focused on comparing the latest portable nanopore sequencing methods with established

Illumina sequencing for vertebrate eDNA metabarcoding of Zambian water samples; moreover, a cost-effective evaluation of different methods of water filtration approaches, and database settings was provided.

The manuscript is well written and organized, although a minimum of revision of the English by a native speaker could improve its overall fluency. The use of figures and tables is appropriate.

The introduction lacks an important premise regarding the effectiveness of the eDNA method in general. The limits connected to the taxonomic assignment of the references present today in the databases are not mentioned, among other things, of little quantity, as evidenced by the authors speaking individually of the studied taxonomic groups. In addition to a quantitative argument, an not non-negligible problem also concerns the fact that the annotations in most public databases are made without any taxonomic revision, and the classical morphological identification of the species can sometimes be doubtful, especially when talking about cryptic or elusive species. The use of eDNA at present does not solve this problem; on the contrary, it probably aggravates it, assigning OTUs as sequences that may not be exactly the species noted. In this light, could the comparison of different sequencing methods and instrumentation have potential implications? Or does it remain a basic limitation of the eDNA technique, for the moment, not unlimited? This discourse should be introduced and argued, with the support of appropriate consequences.

The contamination of negative samples is reported in the results and briefly discussed. At present, it is not very clear what this contamination may be due to, and the authors do not explain well how the repetitive presence of such contamination in a significant number of samples influenced the study, casting doubt on the potential bias of the same.

It is not clear from the tables in the results sections if this contamination was equal for both extraction methods; this could be an interesting data to show and argue better. The effect that this contamination may have had on the study, and the authors' idea regarding its origin, should be explained and better argued.

Another important limitation of the study concerns the comparative methods of analysis of aquatic diversity, mainly fish, which are excluded from most of the effects of the study, and trapping is used as a method to normalize data from eDNA monitoring. But what are the standard methods for monitoring these organisms in the studied area? This is not mentioned, as well as the decision not to apply these methods to make the effects of the study wider. From this point of view, it is a pity that we cannot be relevant to an abundance of vertebrate species.

Best regards

Reviewer #3: This manuscript presents a timely and relevant evaluation of nanopore sequencing as a tool for environmental DNA (eDNA)-based biodiversity monitoring in the Luangwa Valley, Zambia. The authors provide a comprehensive comparison of nanopore and Illumina platforms, exploring aspects such as cost-effective filtration methods, taxonomic assignment strategies, and field applicability. The methodological framework is robust and responds to current needs for accessible genomic tools in low- and middle-income countries.

However, while technically rigorous, the manuscript does not sufficiently articulate the ecological significance of its findings. The framing remains primarily methodological, with limited integration of ecological context, species behavior, or environmental gradients. To increase the impact and clarity of the manuscript (especially for an audience concerned with freshwater biodiversity dynamics) major revisions are needed to anchor the research in ecological theory and regional ecosystem understanding. The discussion should go beyond sequence performance metrics to explore how findings inform our knowledge of biodiversity, species distribution, and ecosystem monitoring practices.

Major comments

Abstract

Your abstract does not highlight any values. You need to highlight the values for better understanding.

Introduction

The current introduction lacks a clearly articulated ecological rationale. The focus is on technical limitations in LMICs without linking these constraints to ecological or conservation challenges specific to African savannah ecosystems.

Recommendation : Reformulate the objectives to explicitly include ecological hypotheses, e.g., “This study investigates whether nanopore sequencing can effectively detect vertebrate biodiversity in dynamic subtropical freshwater systems, with implications for ecological monitoring in seasonally variable habitats.” Introduce the concept of ecological detectability (species behavior, habitat use, waterbody permanence) and refer to studies that illustrate biodiversity dynamics in similar African wetland systems.

Materials and methods

Although the sampling locations are georeferenced and categorized (ponds vs river), there is no ecological description of these habitats. Please add a descriptive map of the study area.

Recommendation: Enrich the section with ecological characterizations of the sites (e.g., temporary vs permanent waterbodies, anthropogenic vs natural origin). These descriptions could frame subsequent interpretations of taxonomic detections, especially considering species habitat preferences and seasonal usage of aquatic environments by terrestrial fauna (e.g., hippos, amphibians).

Discussion

The list and frequency of species detected are reported but not discussed in relation to their ecological traits or habitat use.

Recommendation : In this section, comment on the ecological roles and behaviors of key detected species (e.g., Hippopotamus amphibius as an aquatic-terrestrial connector, Xenopus species as indicators of water quality). Highlight how eDNA reveals cryptic taxa that may not be easily detected through camera trapping, especially amphibians and small mammals.

Minor comments

Lines 22–26, 29 and 76–80 : Remove the personal pronouns from the sentences.

Recommendation : For example, Line 22: This study compares the latest portable nanopore sequencing methods...

Lines 70–75 : This is a very long sentence that is difficult to understand.

Recommendation : Split the text into two sentences.

Reviewer #4: The study titled “Evaluation of nanopore sequencing for increasing accessibility of eDNA studies in biodiverse countries” was aimed at comparing the latest portable nanopore sequencing methods with established Illumina sequencing for vertebrate eDNA metabarcoding of Zambian water samples. A cost-effective versus established water filtration approaches was evaluated, and contrasted with a comprehensive, computationally intensive taxonomic database search with a streamlined manually curated database search. The study was well designed and thought out. I have appended below comments that may help improve the manuscript:

The introduction was well written, however there seem to be so many wordy sentences as most of the sentences were difficult to understands, I suggest breaking them up into sizeable texts that can easily be comprehended. Other few comments can be seen in the attached document.

The materials and methods section was well written and structured, but I was expecting to a subsection dedicated to description of location and even the entire study area. I didn’t see map of the sampled locations; this is key in such ecological studies.

The results are okay; however, I found it difficult to see information on the figures clearly, this need to be recreated for visibility.

The discussion is okay, but there seem to be too much unnecessary information which made the discussion in some part redundant, so I would suggest you revisit this section and objectively discuss your key findings and reduce the excessive explanation of none relevant terms or concepts with regards to the topic of discussion.

**Do you want your identity to be public for this peer review?** For information about this choice, including consent withdrawal, please see our Privacy Policy

Reviewer #1: **Yes: ** Zubia Masood

Reviewer #2: No

Reviewer #3: No

Reviewer #4: **Yes: ** Prof. Ovie Edegbene

---

## [Author Response · Author response to Decision Letter 1]

5 Sep 2025

Dear Editor and Reviewers,

We would like to thank you for your constructive and insightful comments on our manuscript “Evaluation of nanopore sequencing for increasing accessibility of eDNA studies in biodiverse countries.” Your feedback has been invaluable in improving the quality and clarity of the manuscript.

Please find attached our “Response to Reviewers” document, in which we address each of your comments and suggestions point by point. We have also highlighted all the corresponding changes made to the manuscript, which we believe have further improved its quality.

Sincerely,

Daniel Gygax

---

## [Decision Letter · Decision Letter 1]

22 Sep 2025

Evaluation of nanopore sequencing for increasing accessibility of eDNA studies in biodiverse countries

PONE-D-25-24113R1

Dear Dr. Gygax,

We’re pleased to inform you that your manuscript has been judged scientifically suitable for publication and will be formally accepted for publication once it meets all outstanding technical requirements.

Kind regards,

Mizanur Rahman, Ph.D.

Academic Editor

PLOS ONE

Additional Editor Comments (optional):

Reviewer #1:

Reviewer #2:

Reviewer #3:

Reviewer #4:

Reviewers' comments:

Reviewer's Responses to Questions

**Comments to the Author**

Reviewer #1: All comments have been addressed

Reviewer #2: All comments have been addressed

Reviewer #3: All comments have been addressed

Reviewer #4: All comments have been addressed

2. Is the manuscript technically sound, and do the data support the conclusions?

Reviewer #1: Yes

Reviewer #2: Yes

Reviewer #3: Yes

Reviewer #4: Yes

3. Has the statistical analysis been performed appropriately and rigorously?

Reviewer #1: Yes

Reviewer #2: Yes

Reviewer #3: Yes

Reviewer #4: Yes

4. Have the authors made all data underlying the findings in their manuscript fully available?

Reviewer #1: Yes

Reviewer #2: Yes

Reviewer #3: Yes

Reviewer #4: No

5. Is the manuscript presented in an intelligible fashion and written in standard English?

Reviewer #1: Yes

Reviewer #2: Yes

Reviewer #3: Yes

Reviewer #4: Yes

Reviewer #1: Current manuscript have been improved after all reviewer suggestions incorporated by the authors in this article.

Reviewer #2: Dear Authors,

thanks to have seriously considered my previous comments on your document, addressing them as possible or highlighting the limitations as well.

Best regards

Reviewer #3: The authors have taken into account the amendments made to the document in order to improve its scientific quality. The document can now be published in PLOS ONE.

Reviewer #4: Dear authors, I have re-assesses your manuscript and all my concerns have been addressed accordingly.

All the best.

**Do you want your identity to be public for this peer review?** For information about this choice, including consent withdrawal, please see our Privacy Policy

Reviewer #1: **Yes: ** Zubia Masood

Reviewer #2: **Yes: ** Marco Albano

Reviewer #3: No

Reviewer #4: **Yes: ** Prof. Augustine Ovie Edegbene

---

## [Editor Report · Acceptance letter]

PONE-D-25-24113R1

PLOS ONE

Dear Dr. Gygax,

I'm pleased to inform you that your manuscript has been deemed suitable for publication in PLOS ONE. Congratulations! Your manuscript is now being handed over to our production team.

Kind regards,

on behalf of

Dr. Mizanur Rahman

Academic Editor

PLOS ONE